# Persistently Elevated HBV Viral-Host Junction DNA in Urine as a Biomarker for Hepatocellular Carcinoma Minimum Residual Disease and Recurrence: A Pilot Study

**DOI:** 10.3390/diagnostics13091537

**Published:** 2023-04-25

**Authors:** Selena Y. Lin, Dina Halegoua-DeMarzio, Peter Block, Yu-Lan Kao, Jesse Civan, Fwu-Shan Shieh, Wei Song, Hie-Won Hann, Ying-Hsiu Su

**Affiliations:** 1JBS Science Inc., Doylestown, PA 18901, USA; 2Department of Medicine, Thomas Jefferson University Hospital, Philadelphia, PA 19107, USA; 3Division of Gastroenterology and Hepatology, Department of Medicine, Thomas Jefferson University Hospital, Philadelphia, PA 19107, USA; 4Department of Translational Science, Baruch S. Blumberg Institute, Doylestown, PA 18901, USA

**Keywords:** HBV integration, minimal residual disease, urine cell-free DNA, hepatocellular carcinoma, recurrence monitoring

## Abstract

Hepatitis B virus (HBV)-host junction sequences (HBV-JSs) has been detected in the urine of patients with HBV infection. This study evaluated HBV-JSs as a marker of minimum residual disease (MRD) and tumor recurrence after treatment in HBV-hepatocellular carcinoma (HCC) patients. Archived serial urine DNA from two HBV–HCC with recurrence as confirmed by MRI and four HBV-related cirrhosis (LC) patients were used. Urinary HBV-JSs were identified by an HBV-targeted NGS assay. Quantitative junction-specific PCR assays were developed to investigate dynamic changes of the most abundant urinary HBV-JS. Abundant urinary HBV-JSs were identified in two cases of tumor recurrence. In case 1, a 78-year-old female with HBV- HCC underwent a follow-up MRI following microwave ablation. While MRI results were variable, the unique HBV-JS DNA, HBV-Chr17, steadily increased from initial diagnosis to HCC recurrence. In case 2, a 74-year-old male with HBV–HCC contained two HBV-JS DNA, HBV-Chr11 and HBV-*TERT*, that steadily increased after initial HCC diagnosis till recurrence. One LC examined had HBV-*TERT* DNA detected, but transiently in 3.5 years during HCC surveillance. HBV-JS DNA was persistently elevated prior to the diagnosis of recurrent HCC, suggesting the potential of urinary HBV-JS DNA to detect MRD and HCC recurrence after treatment.

## 1. Introduction

Liver cancer is the sixth most commonly diagnosed cancer and the third most common cause of cancer death worldwide, with an estimated 900,000 new liver cancer cases and 830,000 deaths during 2020 [1]. Hepatocellular carcinoma (HCC), a major form of liver cancer, has a dismal 5-year mortality rate of 85%, mainly due to late detection and a high recurrence rate [2,3,4,5,6]. Recurrence rates range from 15% for liver transplantation to nearly 100% for surgery or ablation [7,8,9,10,11,12,13]. The high HCC recurrence rate can be attributed to (1) incomplete treatment where minimum residual disease (MRD) remains; (2) micro-metastases due to MRD within the liver; and (3) de novo lesions [6,14]. Thus, recurrence monitoring is mandatory after HCC treatment.

Early Detection of MRD is important for disease management to improve prognosis. Currently, there are no useful postoperative recurrence markers or specific guidelines for how HCC recurrence should be monitored. It has been difficult to detect recurrent HCC and MRD using alpha-fetoprotein (AFP) and desgama–carboxy prothrombin (DCP) measurements which have limited sensitivity and specificity in patients with non-tumorous hepatic disorders and post-ablation due to active hepatic necrosis that can result in increased levels [15] and by serial imaging [8,9,10,16,17,18,19]. The gold standard for HCC diagnosis is magnetic resonance imaging (MRI)/computed tomography (CT) imaging but is limited in its ability to detect small tumors (<2 cm), tumors in the presence of previously treated lesions (especially from local ablation), cirrhosis, obesity, and dysplastic nodules [9,10,16]. Thus, there is an urgent unmet medical need to have a sensitive test for the detection of MRD and monitoring of HCC recurrence.

Cell-free tumor DNA (ctDNA), the DNA derived from apoptotic tumor cells released into circulation from tumors, is a promising tumor marker with versatile applications, including monitoring tumor progression or resistance to chemotherapies [20,21,22,23]. We have demonstrated that fragmented cell-free DNA (cfDNA) in urine contains DNA derived from solid tumors, including HCC and colon cancer if such a tumor is present [24,25,26,27,28,29]. By developing assays tailored for short fragmented urine ctDNA (uctDNA) for detecting HCC-associated DNA modifications, we have demonstrated the promise of urine for (1) the effective monitoring for HCC recurrence in a pilot study [30], (2) sensitive measurement of MRD after surgical resection [27], and (3) detection of 30% more HCC as compared to serum AFP alone in a 609-subject study [31].

Hepatitis B virus (HBV) is the leading cause of liver cancer, contributing to 60% of HCC in Africa and East Asia. Integrated HBV DNA has been shown to be in 80–90% of HBV-related HCC [32,33,34,35]. HBV-host junction sequences (HBV-JSs) containing both host and viral sequences serve as markers for infected hepatocytes and thus were used to trace the clonal expansion of HCC [36,37] and emerge as a potential marker for HCC recurrence [38]. Recently tumor-identified HBV-JS have been detected in plasma as a potential ctDNA marker for detecting recurrence [39] and in urine by us [29] in patients with HBV infection, including HBV–HCC. It has been suggested immune surveillance eliminates premalignant or malignant cells before they can develop into detectable tumors [40]. Consequently, it is necessary to monitor dynamic marker levels or marker persistency in liquid biopsy to distinguish a marker’s basal noise level due to immune surveillance from clinically relevant marker detection levels. This six-case study aims to provide pilot evidence to test our hypothesis that persistently elevated urine HBV-JS DNA can serve as a biomarker for HBV–HCC MRD and recurrence monitoring.

## 2. Case Presentations

### 2.1. Case Selection

To evaluate the urinary HBV-JS DNA as a potential indicator of minimum residual disease (MRD) and for monitoring tumor recurrence in patients with HBV–HCC, we first identified HBV–HCC cases that met the following criteria for investigation: (1) having HCC recurrence confirmed by MRI, (2) with at least four available serial archived urine cell-free DNA (ucfDNA) with one serial DNA sample containing at least 20 ng DNA that could be used for HBV-targeted NGS assay for HBV-JS detection as previously described [29], and (3) with at least 10 copies of HBV DNA per mL urine as measured by HBV DR2 qPCR assay (JBS Science Inc., Doylestown, PA, USA). Two recurrent HBV–HCC cases, cases 1 and 2 were identified (Table 1). One archived ucfDNA from each case, the first recurrence for case 1 and 3 months prior to recurrence for case 2, with more than 20 ng ucfDNA, were selected for HBV-targeted NGS assay for HBV-JS detection. Four HBV-related cirrhosis (LC) patients whose urine samples were collected under HCC surveillances that had at least 4 serial archived DNA available were selected as negative controls for HCC, listed as cases 3–6 in Table 1.

### 2.2. Case 1

A 78-year-old female with HBV-related cirrhosis was diagnosed with HCC in 2015. After microwave ablation, the patient was monitored for recurrence by MRI and serum AFP. Follow-up MRI revealed a new LI-RADS 3 lesion 1 year later with serum AFP of 2.65 ng/mL. Subsequent imaging remained radiographically indeterminate until 2018, when the lesion was classified as definite HCC (LI-RADS 5).

The last available archived urine DNA for HBV-JS NGS analysis was collected over 2 years (8/2016) before the MRI diagnosis of recurrence which was in 2018. We detected one HBV-JS integrated in Chromosome 17 non-coding region (HBV-Chr17) with ≥2 SR (Table 2). To assess the dynamics of HBV-Chr17 over time, a short amplicon junction-specific quantitative PCR (<60 bp) was developed as detailed in Appendix A for the limit of detection, sensitivity and specificity, and primer sequences. We calculated the percent of HBV-Chr17 per host Chromosome 1 (Chr1) copies and plotted with serum AFP level (ng/mL) with the time of urine collection, as shown in Figure 1. The details of the six clinical time points are summarized in Table 3 with HCC treatment history.

Interestingly, while serial AFP levels were negligible and MRI results variable, the unique HBV-JS DNA, HBV-Chr17, steadily increased from initial diagnosis of 1.2% in 2015 to 3.0% and then 7.0% in August 2016, prior to diagnosis of HCC recurrence in October 2018. There was no urine collected when HCC was diagnosed. HBV-Chr17 was detected right after treatment and steadily elevated, suggesting the presence of MRD and suspicion of HCC recurrence from the same clone that contained HBV-Chr17. The patient passed away shortly.

### 2.3. Case 2

A 74-year-old male with HBV-related cirrhosis was diagnosed with HCC in 2014, which recurred with an LI-RADS 5 lesion after 1 year despite loco-regional therapy.

The last available archived ucfDNA used for HBV-JS detection was 3 months before recurrence diagnosis by MRI. Four HBV-JS with ≥2 SRs were detected from the urine specimen collected on 4/2015. Of which, the three most abundant HBV-JSs are integrations in non-coding sequences in Chromosome 11 (HBV-Chr11) with 63 SRs, *TERT* gene (HBV-TERT_1) with 22 SRs, and *GCSHP2* (HBV-GCSHP2) with 3 SRs as listed in Table 2. The fourth HBV-JS is a second *TERT* integration (HBV-TERT_2) with 2 SRs. Note this less abundant HBV-TERT_2 contains a different host-viral integration breakpoint from HBV-TERT_1.

Our hypothesis is that the tumor-derived viral-host junction DNA should be the most abundant among HBV-JSs detected in urine because of clonal expansion in tumorigenesis. Interestingly, HBV-Chr11 and HBV-TERT_1 junctions are much more abundant than HBV-GCSHP2, suggesting HBV-Chr11 and HBV-TERT_1 junctions might be HCC-derived. We thus selected HBV-Chr11 and HBV-TERT_1 for subsequent study of biomarker dynamics in serially collected urine, with clinical time points and HCC treatment detailed in Table 3 and HBV-JS qPCR assays (HBV-Chr11 and HBV-*TERT*) detailed in Appendix A.

As shown in Figure 1, these two junction DNA were detectable in urine collected right after microwave ablation on 11/2014, suggesting the existence of MRD that was not detected by CT scan. Both HBV-Chr11 and HBV-TERT_1 junctions steadily increased from ~10% to 60% and over 100% three months before diagnosis of recurrence by MRI on 7/2015. There was no urine collected when HCC was diagnosed, and the patient passed away shortly after the recurrence diagnosis.

### 2.4. Case 3

A 53-year-old male with HBV-related cirrhosis was under HCC surveillance. Urine specimens were collected from each annual visit. In the urine specimen collected on 5/2015, a frequently recurrent HCC-associated HBV-JS, HBV-*TERT* junction was identified with 9 SRs that were identical to HBV-TERT_1 found in case 2 (Table 2). Interestingly, the detection of HBV-*TERT,* although initially highly abundant in May 2015, was only transiently detected in the surveillance period of 3.5 years. There were no detectable HBV-*TERT* junctions detected in the other six archived urine DNA collected 2 years before 2015 and 2 years after 2015 (Figure 1). No evidence of HCC was suggested by ultrasound, and AFP levels were negligible under 2 ng/mL.

### 2.5. Case 4–6

Of the three remaining LC cases examined, HBV-JSs were detected in two LC archived ucfDNA and were integrations in repeat regions (Table 2), similar to what we previously reported in chronic HBV patients [29]. We have shown most of the HBV-JSs identified in non-HCC chronic HBV-infected patients were found in repeat regions. HBV-JS identified in repeat regions in LC cases were not included in serial monitoring. Thus, in cases 4–6, none of the identified HBV-JS were included in biomarker dynamic analysis.

## 3. Discussion

For the first time, this pilot study demonstrates the potential applicability of using unique urinary HBV-JS DNA as noninvasive biomarkers for the detection of MRD and for monitoring HCC recurrence in patients with HBV–HCC. In this small 6-case study with two MRI-identified recurrent HCC cases, urinary DNA markers were found to be elevated in urine samples as early as 3 years before MRI confirmation. In other words, the detection of HCC-derived HBV-JS DNA after treatment can indicate the existence of MRD, similar to the detection of *CTNNB1* mutation in the urine collected after surgical treatment in our previous study [27]. Interestingly, all three cases selected for HBV-JS assessment were low in AFP, ranging from undetectable to less than 2 ng/mL throughout the study including at the time of recurrence diagnosis by MRI. By assessing the dynamic of HBV-JS in urine collected from cirrhotic patients under HCC surveillance (case 3), we observed a transient elevation of HCC-associated recurrent *TERT* integration, demonstrating that marker dynamic or persistency is needed to separate a noisy level of biomarkers from a clinically relevant marker detection in a liquid biopsy. This highlights the emerging biomarker application of urinary HBV-JS for HCC MRD and tumor recurrence monitoring.

In case 2, the three most frequent HBV-JS were listed from four HBV-JS identified to have at least 2 SRs. Two, HBV-Chr11 and HBV-TERT_1, have much higher SRs, 63 and 22, than the third one, HBV-GCSHP2, which has 3 SRs. We selected the most abundant HBV-JS detected from urine as a biomarker to monitor for recurrence because we believe that the tumor-derived viral-host junction DNA should be the most abundant among HBV-JS detected in urine because of clonal expansion in tumorigenesis. Interestingly, these two HBV-JS DNA were persistently elevated with the time of study till recurrence, suggesting they were correlated with tumor growth and are HCC-derived HBV-JS.

Although this is a small longitudinal pilot study, the potential of HBV-JS DNA markers for the management of HCC recurrence and detection of MRD is demonstrated. First, for both recurrent cases (cases 1 and 2), HBV-JS DNA markers were elevated before and at the time of diagnosis by MRI imaging. MRI/CT imaging is the gold standard for diagnosing recurrent HCC, but its ability to detect early recurrence, especially after local ablation in the previously treated areas, is limited. This may explain why the DNA markers were found in urine earlier than the MRI diagnosis. Secondly, HCC, like other solid cancers, is a disease of the genome. Detection of HCC genetics should provide not only sensitive and earlier detection for monitoring HCC recurrence and unambiguous assessment for MRD and may also provide HCC genetic information to assist in patient management. Integrated HBV DNA, found in >85% of HBV–HCC, can play a significant role in HBV-related liver disease progression. An HBV-JS created through each integration event provides a molecular signature for each infected hepatocyte and has been used to determine HBV–HCC clonality. Its high abundance can be used as a biomarker for clonal expansion.

Interestingly, in this pilot study of six patients, we detected two identical HBV-*TERT* junctions, the most frequent recurrently targeted gene by HBV integration found in HCC patients [41]. The majority of *TERT* junction coordinates should be unique as HBV integration is through a non-sequence-specific mechanism. However, in our previous study, we found that 76% of *TERT* integrations (n = 477) mainly occur in a small 15 kb region upstream of exon 1, and within that region, over 50% were found in the ~1kb promoter region (Chr5:1,295,162–1,296,162). This small window of frequent *TERT* integration contained 10% of HBV-*TERT* integrations that shared identical viral-host breakpoints. It may not be surprising that the HBV-*TERT* junction identified in case 2 and case 3 of this study coincided with the previously reported repeat HBV-*TERT* integration.

As we’ve demonstrated previously [29] that HBV-JS DNA can be detected in patients with HBV infection, including hepatitis and LC, we also detected HBV-JS DNA in three of four LC patients. Although integration was found in a known HCC-driver gene, *TERT*, for case 3 LC, it was only detected transiently. This could be due to the interplay of the host immune system, tumorigenesis, and the existence of HBV DNA integration events, as we have previously suggested [42]. Here, nucleos(t)ide analogs were effective in reducing viral replication as indicated by low viral serum load (Table 1) but were not able to eliminate existing integrated HBV DNA in infected hepatocytes. HBV-JSs represent a unique biomarker that can serve as a marker for existing HBV infection as well as a liver-specific marker detectable in urine due to apoptosis of infected hepatocytes. Interestingly, a known HCC-associated recurrent *TERT* integration was detected in urine from an LC patient (case 3). It is possible that the *TERT* integration was derived from either neo-antigen expression or indicative of tumorigenesis. Likely, its elimination by the host immune system resulted in a spike of detectable HBV-*TERT* junction in urine. On the other hand, detection of persistently elevated HBV-JS marker values reflects persistent tumor growth whereby some tumor cells that harbored HCC-derived junctions were eliminated by apoptosis, thus serving as an important biomarker for HCC MRD and recurrence monitoring, as schematically shown in Figure 2.

Although this study is limited by the small cohort size, it provides feasibility evidence that detection of highly abundant HBV-JS in urine from patients with HBV–HCC (for example, HBV-Chr11 and HBV-TERT_1 in case 2) could be HCC-derived and served as an HCC marker for monitoring HCC recurrence and MRD assessment in a noninvasive urine liquid biopsy. Furthermore, since the collection of urine can be done at home and then shipped to certified laboratories for testing, the urine test can result in better compliance without the need for a doctor’s office visit.

It is important to note that the levels of urine DNA markers fluctuate for several reasons, including the hydration of the patient at the time of collection, which can result in diluted DNA in the urine. Further studies are needed to identify a proper internal control for this work, which is currently in progress.

Despite the lack of FDA-approved HCC therapies since Sorafenib, the first FDA-approved HCC chemotherapy over 10 years ago, the recent approval of the immune checkpoint inhibitors (ICIs) PD-L1 inhibitor atezolizumab plus the antiangiogenic agent bevacizumab as front-line treatment for advanced HCC and the IMbrave 150 phase III trial has established a novel standard of care in a relatively short period of time. Despite this unprecedented treatment paradigm shift in HCC treatment, only a proportion of HCC patients benefit from immunotherapy, highlighting the need for biomarkers for assessing MRD and predicting treatment response. Encouragingly, the advent of molecular profiling has produced notable promise in the management of HCC associated with response to ICIs. Genomic aberrations such as Wnt/β-catenin mutations in HCCs have been reported to be significantly associated with resistance to immunotherapy compared to wild-type patients, emerging as potential predictive biomarkers of immunotherapy. The HCC medical community is called to put more effort aimed at evaluating novel biomarkers of response to ICIs, considering tumor-intrinsic (e.g., PD-L1 expression, tumor mutation burden (TMB), microsatellite instability (MSI) status, etc.), immune-specific, and combinatorial biomarkers [43,44,45]. The identification of predictive biomarkers of response to immunotherapy remains a priority in HCC, especially considering that the number of indications and patients receiving ICIs is expected to increase in the near future. Thus, it is critical to identify more genetic markers in addition to PD-L1 expression, TMB, and MSI status to select the right patients to receive immunotherapy and to provide useful information for disease monitoring and treatment-decision making. HBV integration is known to cause insertional mutagenesis of HCC-associated genes and induce chromosomal instability [41]. The mutagenesis resulting from HBV integration should be included in the TMB load and evaluated for its impact on disease management. Although this is a pilot study, the detection of HCC-derived urinary HBV-JS as a biomarker for assessing treatment efficacy or MRD and its persistently increasing levels prior to recurrence is encouraging and consistent with our previous β-catenin mutation study in the urine of HCC patients with recurrence after surgical removal of primary HCC [27]. Together, a large cohort study is warranted to comprehensively determine and develop these potential biomarkers for assessing treatment efficacy, MRD, recurrence, and predicting treatment response.

In conclusion, we have demonstrated the potential of urinary HBV-JS DNA for the early detection of HCC recurrence and MRD assessment. Furthermore, we concluded that marker dynamics, such as persistent elevations of a highly abundant HBV-JS DNA, are important for the clinical relevance of HCC detection.

## Figures and Tables

**Figure 1 diagnostics-13-01537-f001:**
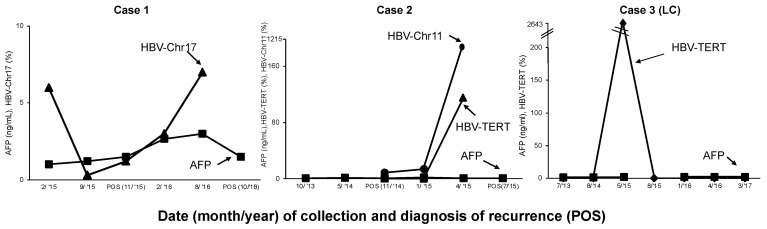
HBV-JS DNA biomarkers levels in serial urine samples from HBV–HCC patients with HCC recurrence and an HBV-cirrhotic patient under HCC surveillance. Both HBV–HCC patients were being monitored for HCC recurrence by MRI and serum AFP. The urine samples were collected prospectively from two HBV–HCC patients (when available) after curative treatment and at follow-up visits and from one cirrhotic patient at HCC surveillance visit. Samples were retrospectively measured for HBV-JS DNA biomarkers. HBV-JS DNA along with serum AFP (ng/mL), were plotted at office visits until the last available visit in which an MRI was performed. The “POS” represents detection of HCC recurrence by MRI. HCC: hepatocellular carcinoma; AFP: alpha-fetoprotein.

**Figure 2 diagnostics-13-01537-f002:**
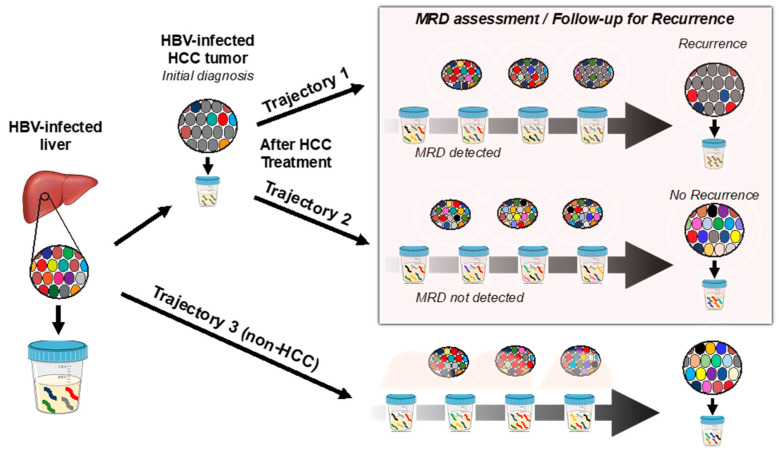
Schematic diagram of urinary HBV-JS detection trajectories during serial sampling of HBV-infected patients undergoing HCC surveillance for primary and recurrent HCC. Each colored circle represents an HBV-infected hepatocyte harboring a unique HBV-host junction sequence (HBV-JS). Gray-colored circles and urinary DNA fragments represent HCC cells and HCC-derived HBV-JS, respectively. Trajectory 1 illustrates an HBV–HCC patient undergoing HCC treatment after an initial HCC diagnosis and then follow-up for HCC recurrence, where serial urine sampling reveals detection of a low amount of HCC-derived HBV-JS (gray-colored DNA) after treatment, suggesting MRD and an increasing level of an HCC-derived HBV-JS detected in urine, up to recurrence diagnosis. Trajectory 2 depicts an HBV–HCC patient whose serial urine sampling reveals a highly diverse collection of HBV-JS after HCC treatment and no detection of original HCC-derived HBV-JS that was detected during the initial HCC diagnosis indicating no MRD nor recurrence. Trajectory 3 represents an HBV-infected non-HCC patient under HCC surveillance with serial urine sampling showing transient levels of HBV-JS detected in urine, as indicated by red DNA.

**Table 1 diagnostics-13-01537-t001:** Clinical characteristics of patient urine for viral-host junction detection by HBV-targeted NGS assay.

Case	Diagnosis	Age	Gender	Antiviral Therapy	ALT (IU/L)	Serum AFP (ng/mL)	HBV Serum (IU/mL)	HBe Ag (−/+)	Hbe Ab	HCC Stage	Tumor Size (cm)
1	HCC	78	F	LAM, TDF	NA	1	ND	-	+	I	0.74
2	HCC	72	M	LAM, TDF	30	1.4	<20	-	-	NA	3.4
3	LC	53	M	LAM, TAF, TDF	27	1.5	<20	ND	ND	NA	
4	LC	68	F	-	NA	9.1	<10	ND	ND		
5	LC	60	M	LAM, ENT	18	3	<20	-	-		
6	LC	57	M	IFN, LAM, TDF	83	1.8	<20	+	-		

LAM, lamivudine; TDF, tenofovir disoproxil fumarate, TAF, tenofovir alafenamide; ENT, entecavir; IFN, pegylated interferon α; NA, not available; ND, not done. Note, all the clinical data were recorded as close as possible to the time of urine collection.

**Table 2 diagnostics-13-01537-t002:** HBV-JS detected in urine by HBV-targeted NGS assay.

Case	Sample Used (Month/Year)	Disease Status	HBV-JS * Gene (Supporting Read)
1	11/2015	3 years pre-recur (HCC)	Chr17_NC (2)
2	4/2015	3 months pre-recur	TERT_1 (22), chr11_NC (63), GCSHP2 (3)
3	5/2015	LC	TERT_1 (9), Chr7_NC (2)
4	7/2019	LC	NA
5	9/2013	LC	Simple repeat (9), LTR (6)
6	7/2015	LC	SINE (17), Simple repeat (6)

* Up to three most abundant HBV-JS listed; NA: no HBV-JS with at least 2 SR detected. LTR: long terminal repeat; SINE: short interspersed nuclear elements.

**Table 3 diagnostics-13-01537-t003:** Clinical characteristics and HCC treatment history over six time points for three cases studied for HBV-JS dynamic.

Case	Data Collection Timepoint (Month/Year)	Follow-Up (Months)	Disease Status (HCC Treatment)	Serum AFP (ng/mL)	LI-RADS	HCC Stage	Tumor Size (cm)
1	2/2015	0	Pre-recur	1.0	NA	NA	NA
	9/2015	7	Pre-recur	1.21	NA	NA	NA
	11/2015	9	POS (MW)	1.21	NA	I	0.74
	2/2016	12	Pre-recur	2.65	3	NA	NA
	8/2016	18	Pre-recur	1.08	4	NA	NA
	10/2018	44	POS	1.5	5	I	1.2
2	10/2013	0	Pre-recur	1.4	NA	NA	NA
	5/2014	7	Pre-recur	1.2	NA	NA	NA
	11/2014	13	Pre-recur	1.2	NA	NA	NA
	1/2015	15	POS (MW)	1.5	5	NA	3.4
	4/2015	18	Pre-recur	1	NA	NA	NA
	7/2015	21	POS (TACE)	1	NA	NA	0.6
3	7/2013	0	Cirrhosis	1.5	NA	NA	NA
	8/2014	14	Cirrhosis	1.5	NA	NA	NA
	5/2015	22	Cirrhosis	1.8	NA	NA	NA
	8/2015	25	Cirrhosis	1.8	NA	NA	NA
	1/2016	31	Cirrhosis	2.1	NA	NA	NA
	4/2016	33	Cirrhosis	1.9	NA	NA	NA
	3/2017	45	Cirrhosis	1.9	NA	NA	NA

Pre-recur indicates timepoint pre-HCC recurrence; POS indicates timepoint at positive HCC recurrence diagnosis; MW, patient underwent microwave ablation; TACE, patient underwent transcatheter arterial chemoembolization.

## Data Availability

The data generated and analyzed during this study are available from the corresponding author upon reasonable request.

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
