# Peer review of "Persistently Elevated HBV Viral-Host Junction DNA in Urine as a Biomarker for Hepatocellular Carcinoma Minimum Residual Disease and Recurrence: A Pilot Study"

_diagnostics, 2023, doi:10.3390/diagnostics13091537_

Round 1

Reviewer 1 Report

Journal: Diagnostics (ISSN 2075-4418)

Manuscript ID:diagnostics-2249829

Type:Article

Title:Persistently elevated HBV viral-host junction DNA in urine as a biomarker for hepatocellular carcinoma minimum residual disease and recurrence

Comments: The authors of this study evaluated HBV-JSs as a marker of minimum residual disease (MRD) and tumor recurrence after treatment in HBV-hepatocellular carcinoma (HCC) patients.Their results suggest that HBV- JS DNA were persistently elevated prior to diagnosis of recurrent HCC suggesting the potential of urinary HBV-JS DNA to detect MRD and HCC recurrence after treatment. The subject of this manuscript is interesting and valuable, but there are a few of defects need to be modified.

1. The author defines Hepatitis B virus (HBV)-host junction sequences (HBV-JSs) in the Introduction section (Paragraph 4: HBV-host junction sequences (HBV-JSs) containing both host and viral sequences serve as......). It is suggested that the author define Hepatitis B virus (HBV)-host junction sequences (HBV-JSs) in the abstract Background section, so that readers can better understand the research content of this article.

2. The abbreviation of HBV-LC is a bit difficult to understand., is it HBV-related liver cirrhosis (LC). 

3. Introduction section: It has been difficult to detect recurrent HCC and MRD using alpha-fetoprotein (AFP) and desgama-carboxy prothrombin (DCP) measurements and serial imaging [8-10,13-15]. Additionally, the gold standard for HCC diagnosis is magnetic resonance imaging (MRI)/computed tomography (CT) imaging......It is suggested to briefly explain the limit of AFP and DCP in detecting recurrent HCC and MRD.

4. Should the author add the treatment method and time point in Table 3, so that the reader can better understand the persistent elevations of a highly abundant HBV-JS DNA in urine after treatment.

Author Response

Response to Reviewer #1 Comments (italicized):

The authors of this study evaluated HBV-JSs as a marker of minimum residual disease (MRD) and tumor recurrence after treatment in HBV-hepatocellular carcinoma (HCC) patients. Their results suggest that HBV- JS DNA were persistently elevated prior to diagnosis of recurrent HCC suggesting the potential of urinary HBV-JS DNA to detect MRD and HCC recurrence after treatment. The subject of this manuscript is interesting and valuable, but there are a few of defects need to be modified.

  1. The author defines Hepatitis B virus (HBV)-host junction sequences (HBV-JSs) in the Introduction section (Paragraph 4: HBV-host junction sequences (HBV-JSs) containing both host and viral sequences serve as......). It is suggested that the author define Hepatitis B virus (HBV)-host junction sequences (HBV-JSs) in the abstract Background section, so that readers can better understand the research content of this article.

Response: Thank you for the suggestion. The HBV-JS is now defined in the revised Abstract on line 21.

  1. The abbreviation of HBV-LC is a bit difficult to understand., is it HBV-related liver cirrhosis (LC). 

Response: We had revised HBV-LC to indicated HBV-related liver cirrhosis in the Abstract and Results section 3.1.

  1. Introduction section: It has been difficult to detect recurrent HCC and MRD using alpha-fetoprotein (AFP) and desgama-carboxy prothrombin (DCP) measurements and serial imaging [8-10,13-15]. Additionally, the gold standard for HCC diagnosis is magnetic resonance imaging (MRI)/computed tomography (CT) imaging......It is suggested to briefly explain the limit of AFP and DCP in detecting recurrent HCC and MRD.

Response: We added a brief explanation in lines 49-51.

  1. Should the author add the treatment method and time point in Table 3, so that the reader can better understand the persistent elevations of a highly abundant HBV-JS DNA in urine after treatment.

Response: The HCC treatments received during urine serial sample collection is now listed in Table 3 for HCC patients in study.

Reviewer 2 Report

This interesting study assesses a current, timely topic in HCC.
We recommend some changes:
- We believe this article is suitable for publication in the journal although some revisions are needed. The main strengths of this paper are that it addresses an interesting and very timely question and provides a clear answer, with some limitations. Certainly, the study is limited to a very small sample size, and the authors should further express this point. Thus, the authors should better highlight the limitations of the current paper.
Immune checkpoint inhibitors (ICIs) including pembrolizumab, nivolumab, durvalumab, atezolizumab, etc. have been recently evaluated in HCC patients, and clinical trials assessing single-agent ICI have reported disappointing results. Conversely, immune-based combinations have been more striking. In fact, the phase III IMbrave150 trial assessing the combination of the antiangiogenic agent bevacizumab plus the PD-L1 inhibitor atezolizumab versus single-agent sorafenib has established a new standard of care for HCC patients with advanced disease. According to IMbrave150, atezolizumab - bevacizumab have reported statistically significant and clinically meaningful benefits in several clinical outcomes, including objective response rate (ORR), progression-free survival (PFS), and overall survival (OS), with these advantages also confirmed by the updated results of this trial, showing a median OS of more than 19 months in HCC patients receiving the immune-based combination. Despite ICI seem to have finally found their role in HCC as part of combinatorial strategies, several questions remain unanswered. Among these, the lack of validated biomarkers of response represents an important issue since only a proportion of HCC patients benefit from immunotherapy. Based on these premises, a greater understanding of the role of potential biomarkers including programmed death ligand 1 (PD-L1) expression, tumor mutational burden (TMB), microsatellite instability (MSI) status, gut microbiota and several others is fundamental. In addition, clinical trials on HCC immunotherapy widely differed in terms of drugs, patients, designs, terms of study phases, and inconsistent clinical outcomes. The background of the changing scenario of medical treatment in HCC should be better discussed, and some recent papers regarding this topic should be included (PMID: 35403533; PMID: 36368251; PMID: 34976841  ).

- Discussion section: Very interesting and timely discussion. Of note, the authors should expand the Discussion section, including a more personal perspective to reflect on. For example, they could answer the following questions – in order to facilitate the understanding of this complex topic to readers: what potential does this study hold? What are the knowledge gaps and how do researchers tackle them? How do you see this area unfolding in the next 5 years? We think it would be extremely interesting for the readers.

Author Response

Response to Reviewer #2 Comments (italicized):

This interesting study assesses a current, timely topic in HCC. We believe this article is suitable for publication in the journal although some revisions are needed. The main strengths of this paper are that it addresses an interesting and very timely question and provides a clear answer, with some limitations.

  1. Certainly, the study is limited to a very small sample size, and the authors should further express this point. Thus, the authors should better highlight the limitations of the current paper.

Response: We have modified the title to reflect the nature of the study, which is a pilot study. We have also revised the discussion to further emphasize the small cohort size and limitations of study as in line 351.

  1. Immune checkpoint inhibitors (ICIs) including pembrolizumab, ….. Among these, the lack of validated biomarkers of response represents an important issue since only a proportion of HCC patients benefit from immunotherapy. Based on these premises, a greater understanding of the role of potential biomarkers including programmed death ligand 1 (PD-L1) expression, tumor mutational burden (TMB), microsatellite instability (MSI) status, gut microbiota and several others is fundamental. In addition, clinical trials on HCC immunotherapy widely differed in terms of drugs, patients, designs, terms of study phases, and inconsistent clinical outcomes. The background of the changing scenario of medical treatment in HCC should be better discussed, and some recent papers regarding this topic should be included (PMID: 35403533; PMID: 36368251; PMID: 34976841).

Response: We thank the reviewer for the elaborate information of recent ICI findings in HCC and the suggestion to include this in the discussion of HCC treatment, and the importance of having validated biomarkers of therapeutic response. We have included this in the revised discussion with citations (Reference #46-48) as suggested (starting from line 328)

  1. Discussion section: Very interesting and timely discussion. Of note, the authors should expand the Discussion section, including a more personal perspective to reflect on. For example, they could answer the following questions – in order to facilitate the understanding of this complex topic to readers: what potential does this study hold? What are the knowledge gaps and how do researchers tackle them? How do you see this area unfolding in the next 5 years? We think it would be extremely interesting for the readers.

Response: We appreciate that the reviewer values our study and have expanded the discussion to include the future outlook and utility of urinary HBV-JS as well as the implications of our findings as included in the response to comment #2.

Reviewer 3 Report

In this manuscript, Lin et al. described that HBV-JS DNA was persistently elevated before a diagnosis of recurrent HCC suggesting the potential of urinary HBV-JS DNA to detect MRD and HCC recurrence after treatment. However, this reviewer has the following concerns. 

Major comments: 

1. First of all, the sample size is too small to evaluate this technology. 

2. In 3.1. Detection of HBV viral-host junction sequence (HBV-JS) in urine, it is described that HCC recurrence is confirmed by MRI, but this reviewer wonders how HCC detected only by ultrasonography is handled.

3. This reviewer wonders why serum DNA was not used for this analysis. Serum DNA would be more advantageous to detect HBV-JS DNA than urine DNA.

4. The reason why HBV-JS DNA was false positive in case 3 is unclear. 

Minor comments: 

1.      In 4. Discussion, page 7, line 8 from the bottom, “HBV-TERT junction identified in case 1 and case 3”. It should be case 2 and case 3.

Author Response

Response to Reviewer #3 Comments (italicized):

In this manuscript, Lin et al. described that HBV-JS DNA was persistently elevated before a diagnosis of recurrent HCC suggesting the potential of urinary HBV-JS DNA to detect MRD and HCC recurrence after treatment. However, this reviewer has the following concerns. 

  1. First of all, the sample size is too small to evaluate this technology.

Response:  We agree the sample size is small and have revised to highlight the nature of this study as a pilot study in the revised title and in discussion (starting on lines 238, 288, and 328) to demonstrate proof-of-concept for using urinary HBV-JSs as a biomarker for HCC MRD and recurrence monitoring and the need for further evaluation of this technology to clinic practice.

  1. In 3.1. Detection of HBV viral-host junction sequence (HBV-JS) in urine, it is described that HCC recurrence is confirmed by MRI, but this reviewer wonders how HCC detected only by ultrasonography is handled.

Response: The detection of recurrence was by MRI imaging as detailed in line 90.

  1. This reviewer wonders why serum DNA was not used for this analysis. Serum DNA would be more advantageous to detect HBV-JS DNA than urine DNA.

Response:  Serum is used for HBV DNA viral load assessment. However, to date serum DNA has not been used for the study of integrated HBV DNA due to the high background of replicating virus, if present, making it challenging to enrich for integrated HBV DNA templates in a background of full-length HBV genome templates. In this study, we use urine DNA as it can be advantageously collected in a noninvasive manner and more importantly is free of active replicating virus as we have previously demonstrated (PMID: 29548283).

  1. The reason why HBV-JS DNA was false positive in case 3 is unclear.

Response: We have expanded the discussion for case 3 to elaborate on the nature of detecting HBV-TERT DNA in a non-HCC patient, lines 288-302. HBV DNA can integrate into the host genome upon infection and its detection in urine of non-HCC patients (HBV-related hepatitis and cirrhosis) has previously been reported by us (PMID: 34558837). In case 3, the transient detection of HBV-JS DNA is not considered false positive, but rather reflects the detection of liver-derived integration in urine.

Point 5.  In 4. Discussion, page 7, line 8 from the bottom, “HBV-TERT junction identified in case 1 and case 3”. It should be case 2 and case 3.

Response:  We have revised accordingly to correctly state case 2 and case 3 (line 283).

Reviewer 4 Report

Authors aimed to evaluate HBV-JSs as a marker of minimum residual disease (MRD) and tumor recurrence  after treatment  in HBV-hepatocellular carcinoma (HCC) patients. 

There are several concerns to be addressed.

1) Since this is a pilot case-control study,it should be addressed in the title.

2) The information of antiral therapy should be described.

3) Whether antiviral therapy itself affects HBV-JSs should be discussed.

4) For the better understanding, the theoretical concept of HBV-JS to detect MRD should be addressed as graphics.

5) Please, cite the related articles; PMID: 35468711/ PMID: 36263666/ PMID: 34814239/ PMID: 34674513/ PMID: 35430783.

Author Response

Response to Reviewer #4 Comments (italicized):

Authors aimed to evaluate HBV-JSs as a marker of minimum residual disease (MRD) and tumor recurrence after treatment in HBV-hepatocellular carcinoma (HCC) patients. There are several concerns to be addressed.

  1. Since this is a pilot case-control study, it should be addressed in the title.

Response: We have revised the title to include this study is a pilot study.

  1. The information of antiviral therapy should be described.

Response:  We had added this information to Table 1.

  1. Whether antiviral therapy itself affects HBV-JSs should be discussed.

Response: We had added this discussion starting from line 288.

  1. For the better understanding, the theoretical concept of HBV-JS to detect MRD should be addressed as graphics.

Response: We have added Figure 2 to schematically show the concept of urinary HBV-JS markers for HCC MRD and recurrence monitoring.

  1. Please, cite the related articles; PMID: 35468711/ PMID: 36263666/ PMID: 34814239/ PMID: 34674513/ PMID: 35430783.

Response: We thank the reviewer for bringing these recent articles to our attention and have included the citations as references 12,13, 19, and 38.

Round 2

Reviewer 2 Report

acceptance

Reviewer 3 Report

Even if it is a pilot study, this reviewer thinks that the sample number is too small to publish in DIagnostics, because urinary HBV-JS DNA was detected in HCC and LC. 

Reviewer 4 Report

Authors addressed raised issues appropriately.